# Effects of Ambient Temperature, Relative Humidity, and Precipitation on Diarrhea Incidence in Surabaya

**DOI:** 10.3390/ijerph20032313

**Published:** 2023-01-28

**Authors:** Bima Sakti Satria Wibawa, Aussie Tahta Maharani, Gerry Andhikaputra, Marsha Savira Agatha Putri, Aditya Prana Iswara, Amir Sapkota, Ayushi Sharma, Arie Dipareza Syafei, Yu-Chun Wang

**Affiliations:** 1Department of Environmental Engineering, College of Engineering, Chung Yuan Christian University, 200 Chung-Pei Road, Zhongli, Taoyuan City 320314, Taiwan; 2Institute of Tropical Disease, Universitas Airlangga, Surabaya 60286, Indonesia; 3Department of Environmental Health, Faculty of Health Science, Universitas Islam Lamongan, Lamongan 62211, Indonesia; 4Department of Civil Engineering, College of Engineering, Chung Yuan Christian University, 200 Chung-Pei Road, Zhongli, Taoyuan City 320314, Taiwan; 5Department of Epidemiology and Biostatistics, University of Maryland School of Public Health, Maryland, MD 20742, USA; 6Department of Environmental Engineering, Institut Teknologi Sepuluh Nopember, Surabaya 60111, Indonesia; 7Research Center for Environmental Changes, Academia Sinica, 128 Academia Road, Section 2, Nankang, Taipei 11529, Taiwan

**Keywords:** diarrhea, generalized additive model, meteorological variables, Surabaya

## Abstract

Background: Diarrhea remains a common infectious disease caused by various risk factors in developing countries. This study investigated the incidence rate and temporal associations between diarrhea and meteorological determinants in five regions of Surabaya, Indonesia. Method: Monthly diarrhea records from local governmental health facilities in Surabaya and monthly means of weather variables, including average temperature, precipitation, and relative humidity from Meteorology, Climatology, and Geophysical Agency were collected from January 2018 to September 2020. The generalized additive model was employed to quantify the time lag association between diarrhea risk and extremely low (5th percentile) and high (95th percentile) monthly weather variations in the north, central, west, south, and east regions of Surabaya (lag of 0–2 months). Result: The average incidence rate for diarrhea was 11.4 per 100,000 during the study period, with a higher incidence during rainy season (November to March) and in East Surabaya. This study showed that the weather condition with the lowest diarrhea risks varied with the region. The diarrhea risks were associated with extremely low and high temperatures, with the highest RR of 5.39 (95% CI 4.61, 6.17) in the east region, with 1 month of lag time following the extreme temperatures. Extremely low relative humidity increased the diarrhea risks in some regions of Surabaya, with the highest risk in the west region at lag 0 (RR = 2.13 (95% CI 1.79, 2.47)). Extremely high precipitation significantly affects the risk of diarrhea in the central region, at 0 months of lag time, with an RR of 3.05 (95% CI 2.09, 4.01). Conclusion: This study identified a high incidence of diarrhea in the rainy season and in the deficient developed regions of Surabaya, providing evidence that weather magnifies the adverse effects of inadequate environmental sanitation. This study suggests the local environmental and health sectors codevelop a weather-based early warning system and improve local sanitation practices as prevention measures in response to increasing risks of infectious diseases.

## 1. Introduction

Globally, diarrhea is the leading cause of mortality and morbidity and is a prominent concern for children younger than five years of age [1,2]. The World Health Organization (WHO) estimated 829,000 deaths annually due to diarrhea in low- and middle-income countries [3]. It is reported that acute infectious diarrhea is a major cause of morbidity among inhabitants of high-income countries, while inhabitants of low- and middle-income countries suffer from a disproportionate burden of diarrhea [4]. United Nations International Children’s Emergency Fund (UNICEF) reported that 4.8% of all deaths among children under age 5 in Indonesia in 2019 were caused by diarrhea [5]. It is known that bacteria, viruses, and parasites are the common infectious agents of diarrhea disease [6]. However, diarrhea disease also can be caused by noninfectious etiology, such as ischemic colitis or inflammatory bowel disease, which can simultaneously present with abdominal pain and bloody diarrhea [7].

Previous studies have noted risk factors closely related to diarrhea diseases, including lack of water, sanitation, and hygiene (WASH) implementation, younger population age, male population, young maternal age, and low maternal education [8,9]. 

Others have pointed out that climatic conditions also play important role in diarrhea disease transmission [10,11]. The National Aeronautics and Space Administration (NASA) has predicted that global temperatures are expected to rise by 2 °C or more by the end of the 21st century [12,13]. Shifting temperature trends can impact infectious disease transmission due to pathogens’ higher replication and inherited adaptation mechanism [14]. In addition, higher rainfall intensity can lead to infrastructure damage, and surface and groundwater supply contamination [15]. Water sources may be contaminated by micro-organisms, such as bacteria (*Salmonella* spp., *Shigella*, *Escherichia coli*, *Campylobacter*, *Vibrio cholera*), viruses (*Rotarovirus*, *Norovirus*, *Adenovirus*), or protozoa (*Giardia*, *Cryptosporidium*, *Cyclospora*), causing gastrointestinal infections during extreme weather events [16]. Higher humidity also affects the replication of bacteria and protozoa since higher humidity promotes the formation of biofilm on surfaces, providing protection and a favorable environment for microorganisms [17,18,19]. Weather variations are also known to influence population behaviors, including eating habits, and may indirectly influence the transmission of this infection [18].

Surabaya is the second-largest city in Indonesia, with a population density of around 9900/km^2^ within the urban area and 2200/km^2^ in the suburban area (total population size was 2.87 million in 2020). A study reported that open defecation is still a problem in Surabaya, with more than 10,000 houses without septic tanks [20]. The limited access to clean water and insufficient wastewater treatment facilities are still common issues in Indonesia [21]. Poor hygiene practice is the root cause of the transmission of diarrhea [22]. Therefore, people under such conditions are easily exposed to unsafe water, increasing the risk of diarrhea infection [23].

The latest IPCC report states that extreme weather event frequency will likely increase in the near future despite the mitigation efforts that have been implemented thus far [24]. It is essential for the public health sector to strengthen the current adaptations and strategies against climate change threats. In terms of diarrhea disease, adaptation and mitigation measures require a better understanding of geographic settings and climate hazards (extremely low and high temperatures, relative humidity, and precipitation). However, there were only a few studies focusing on the diarrhea risk linked with extreme weather patterns in global south countries or metropolitan areas in Indonesia. Therefore, this study investigated the effects of average temperature, precipitation, and relative humidity on the risk of diarrhea in the north, central, west, south, and east regions in Surabaya. The spatial-temporal pattern of diarrhea disease might help identify high-risk locations and may assist in better allocating health care resources for prevention and control.

## 2. Materials and Methods

Surabaya is a metropolitan city (32,636 km^2^), located at latitude −7°14′57.01″ S and longitude 112°45′2.9″ E in East Java, Indonesia (Figure 1A), with more than 2.87 million inhabitants, according to the 2020 census [25]. This study divided Surabaya into five regions, namely, the north, central, west, south, and east regions (Figure 1B). Figure 1C displays the natural geographical information of Surabaya; the east region has the highest ratios of pond area (33.9%) and wet forest (6.72%), followed by the west region, with corresponding ratios of 11.2% and 0.86%.

### 2.1. Data Sources

We retrieved monthly surveillance diarrhea data from January 2018 to September 2020 from six governmental public health centers in Surabaya, namely, the government public health centers of Puskesmas Dr. Soetomo, Asemrowo, Keputih, Siwalankerto, Rungkut, and Simomulyo (Figure 1B). We decided to divide Surabaya City into the aforementioned regions. Health outcome was collected as an aggregate count and all personal information of the insured population was stripped, assuring patient privacy.

Meteorology, Climatology, and Geophysical Agency (Indonesian: Badan Meteorologi, Klimatologi, dan Geofisika, abbreviated: BMKG) provided daily weather data of average temperature (°C), precipitation (mm), and relative humidity (%) for the same period. We aggregated the daily weather data into monthly average weather data from three real-time monitoring stations in Surabaya—Perak I, Maritim Tanjung Perak I, and Juanda (shown in Figure 1B)—that are surface meteorological observatories, providing data from January 2018 to September 2020. Detailed station information and quality assurance criteria are available online (https://dataonline.bmkg.go.id/ accessed on 8 November 2021).

### 2.2. Data Analysis

#### 2.2.1. Incidence Rate Analysis

The incidence rate is the number of individuals who have a particular disease in a population at a certain time, and is represented as cases per 100,000 population [26]:(1)$$Incidence\;rate=\frac{\mathrm{number}\;\mathrm{of}\;\mathrm{new}\;\mathrm{cases}\;\mathrm{at}\;\mathrm{the}\;\mathrm{time}\;\left(\mathrm{or}\;\mathrm{period}\right)}{\mathrm{the}\;\mathrm{average}\;\mathrm{number}\;\mathrm{of}\;\mathrm{people}\;\mathrm{in}\;\mathrm{the}\;\mathrm{same}\;\mathrm{period}}\times 100,000$$

This study calculated the incidence rate by month and region to determine the seasonal (rainy season: November to March; dry season: April to October) and spatial difference in incidence rate from January 2018 to September 2020. Location of study area, weather stations, prevalence rate of diarrhea per 100,000 population, and land use map of Surabaya were plotted using ArcGIS Pro version 3.0.3.

#### 2.2.2. Time-Series Analysis of Diarrhea

To match the temporal resolution between the exposure and outcome variables, we calculated all regional daily meteorological on a monthly term to align it with health data. This study conducted the analysis using generalized additive model (GAM) to maximize the quality of prediction of a dependent variable Y from various distributions by estimating unspecific (non-parametric) functions of the predictor variables that are “connected” to the dependent variable via a link function [27]. The effects of monthly average temperature, precipitation, and relative humidity on Surabaya’s diarrhea disease were evaluated using GAM. Previous studies have proved GAM was very useful in explaining the association between diarrhea and weather determinants [28,29]. The GAM model is as follows:(2)$$Ln\left(Yt\right)=\sum_{i=1}^{I}s\left(x_{it},df\right)+s\left(sin\left(\frac{2\pi t}{12}\right),df\right)+s\left(time,\;df\right)$$
where (*Yt)* denotes the region-specific diarrhea cases in month t; *s*(.) denotes smooth function; and *x_i_* denotes the monthly average temperature, precipitation, and relative humidity. We set degree of freedom (*df*) to 5 for the weather factors. The time lag associations between monthly weather measurements and incidence of diarrhea were evaluated for time lags of 0 to 2 months. A sinusoidal term, sin (2πt/12), was incorporated into the models to control for the season [30], and time was used to control the long-term trend, with the *df* of 7 per year. This study verified the correlation between weather variables using Spearman rank correlation and variance inflation factor (VIF) of weather variables to explain the multicollinearity between variables before model fitting. The final model was tested separately for monthly precipitation and relative humidity adjusted with monthly average temperature to prevent multicollinearity between precipitation and relative humidity. We tried *df* of 3 to 6 for meteorological variables and 3 to 7 for time. The model selection was based on the Akaike information criterion (AIC) value, in which a lower AIC value indicates a better-fitting model [31,32]. The relative risk (RR) and 95% confidence interval (95% CI) were estimated at extremely low (5th percentile) and high (95th percentile) values of monthly average temperature, precipitation, and relative humidity, compared with the measurement with the lowest risk. This study adopted “*mgcv*” and “*gamRR*” in R Software version 1.4.1717 to fit all models.

## 3. Results

### 3.1. Trends of Climatic Characteristics and Diarrhea Morbidity in Surabaya

Figure 1B displays the incidence rate of diarrhea by region during the study period. The total incidence rates (per 100,000 population) were 138 in the north region, 195 in the central region, 484 in the west region, 355 in the south region, and 709 in the east region. The monthly trends of diarrhea cases per 100,000 population, from January 2018 to September 2020, by regions of Surabaya, is presented in Figure 2. The east and south regions reported a similar spike of diarrhea case numbers in 2020, suggesting some underlying environmental risk factors.

Table 1 lists the descriptive statistics of monthly diarrhea cases by study region, and Table 2 lists incidence rates by month; higher incidence rates occurred in the rainy season, i.e., from November to March. The highest incidence rates occurred in January. These rates were 2.41 to 4.42 times the rates in the dry season (April to October) (the numbers were calculated by highest rate divided by lowest rates).

Figure 3 shows the trends of monthly average temperature, precipitation, and relative humidity. The mean monthly average temperature was 28.7 °C in Surabaya, with mean precipitation of 155 mm and mean relative humidity of 75.2% (Table 1).

### 3.2. The Association between Diarrhea Cases and Weather Variables

The Spearman rank correlations between each weather variable (average temperature, precipitation, and relative humidity) are shown in Appendix A. We observed a high correlation between monthly precipitation and relative humidity, with VIF values of 5.13 and 4.88 (*p*-value = 0.02| R^2^ = 0.08) for precipitation and relative humidity, respectively, as we included both covariates in the model. The VIF values of precipitation and relative humidity were both higher than the suggested threshold value of 3.33 (Appendix A) [33]; thus, to prevent multicollinearity, the risk association model was tested separately for monthly precipitation, relative humidity, and monthly average temperature.

Figure 4 displays associations between diarrhea risk and (a) average temperature and (b) relative humidity, by lag time measured in months, in regions of Surabaya. Appendix A list region-specific relative risk (95% confidence interval) of diarrhea associated with the average temperature, relative humidity, and precipitation at the 5th and 95th percentiles, relative to lowest risk measurement at lags 0–2. The lowest risk of diarrhea associated with a weather variable was 28.7 °C for temperature, 75.2% for relative humidity, and 155 mm for precipitation. The diarrhea risks were associated with extremely low and high temperatures in all regions of Surabaya. The highest relative risk of diarrhea associated with extremely low temperature was observed in the east region, after 0 months of lag time, with an RR of 3.22 (95% CI: 2.71, 3.74), and extremely high temperature was found in the east region, 1 month of lag time, with an RR of 5.39 (95% CI: 4.61, 6.17) (Appendix A).

The time lag associations between the risk of diarrhea and extremely low relative humidity was found in north and central regions of Surabaya (Figure 4b). The diarrhea risk associated with extremely low relative humidity was found to be highest in the north region at 2 months of lag time, with an RR of 1.66 (95% CI: 1.51, 1.81). The risk associated with the extremely high relative humidity were significant in the north, central, and west regions of Surabaya. The risk is highest in the west region, after a lag time of 2 months, with an RR of 2.13 (95% CI: 1.79, 2.47).

Appendix A shows that the diarrhea risk is significantly associated with the extremely high precipitation in central region at lag 0 with an RR of 3.05 (95% CI: 2.09, 4.01). Moreover, the extremely low precipitation also associated with diarrhea risk in the north region, after a lag time of 2 months, with an RR of 1.13 (95% CI: 1.02, 1.24) (Appendix A).

## 4. Discussion

Climate change has worsened and intensified the frequency of climate hazards, and this trend is projected to continue in the foreseeable future; therefore, enhancing public health mitigation efforts is an urgent necessity. This study intended to evaluate the spatial and temporal association between the risk of diarrhea and extreme weather conditions (i.e., the 5th and 95th percentile measurements of average temperature, precipitation, and relative humidity) in five regions of Surabaya, Indonesia. Results of GAM models showed that extremely low and extremely high temperatures were associated with increased risk of diarrhea in regions of Surabaya. Meanwhile, extremely high relative humidity significantly affected diarrhea risk in Surabaya. Moreover, extremely high precipitation has a stronger impact on diarrhea risk than extremely low precipitation at lag time of 0 months. This study provides scientific evidence that in a city located in a tropical area, with low variations in monthly weather conditions, common diarrhea disease can still be influenced by extreme weather conditions. Therefore, the potential public health impacts from changing climate and increasing extreme weather events should be carefully evaluated for populations in tropical areas.

Previous studies indicated that unsafe drinking water, poor sanitation, and unhygienic practices were the main causes of the increased morbidity and mortality of diarrhea [34,35]. In addition, scientific evidence suggests that diarrhea is a climate-sensitive disease [11,16,28]. In line with our findings, a study from Brisbane reported an association between extremely low temperatures and increased emergency department visits for childhood diarrhea by 10% (95% CI: 3–18%) [36]. Our results showed that the risk of extremely low temperature-related diarrheal disease was apparent at a lag time of 0 months. Likewise, a study in Taiwan reported that low temperature was associated with the risk of viral diarrhea, with a lag time of up to 3 weeks, with the highest risk occurring 2 weeks after the low temperatures (RR: 1.41, 95% CI: 1.24–1.60) [10]. The extremely low temperature could enhance the replication and survival of viruses; hence, the incidence of viral diarrhea might be increased [36]. A study pointed out that rotavirus follows a similar pattern of childhood viruses and measles, which are transmitted through the respiratory route in low temperature [37]. Another study found that rotavirus is more stable in low temperature conditions compared to high temperature conditions [38].

On the contrary, the present study indicated that higher temperature could increase the risk of diarrhea in the west, south, and east regions of Surabaya. Similar findings are reported from Peru, where an increase of a 1 °C in the mean temperature elevated the clinic visits of childhood diarrhea sufferers by 3.8% (incidence rate ratio (IRR): 1.038, 95% CI: 1.032–1.044) [14]. Moreover, a diarrhea study of the population under 5 years old reported that temperature has a positive association of 0.0497% in districts of Iran (IRR: 1.0497, 95% credible interval (CrI): 1.0254–1.0748) [39]. A possible explanation is that warm ambient environment could speed up the replication of pathogens and may lead to food spoiling easily [35]. In addition, human behaviors may change during hot weather conditions; this may include frequent water consumption, higher reliance on unsafe drinking water sources, and poor hygiene practices that could increase the risks of diarrhea [40]. Prior studies have reported the association between temperature and the risk of enteric disease at different lag times ranging from 0 to 14 weeks [10,41]. A study in Taiwan indicated that extremely hot temperature showed apparent effects on diarrhea disease, at a lag time of 8 weeks (2 months), for bacterial diarrhea (RR: 1.07, 95 % CI: 1.02–1.13) [10]. In line with our findings, we found the effects of high temperature on diarrhea incidence at a lag time of 1 and 2 months, varying by region. The prolonged effects of diarrhea could be caused by the inadequate treatment of diarrhea and the reinfection of cases [42].

A study in Indonesia found rotavirus infection increased during the rainy season [43]. Likewise, a recent study in Taiwan reported that extremely high precipitation (290 mm) elevated the risk of bacterial diarrhea among younger population (RR: 2.77, 95% CI: 1.60–4.76) at a lag time of 8 weeks (2 months) [10]. Similar to previous studies, we found that higher precipitation was significantly associated with increased diarrhea risk in the south Surabaya region. This might be caused by excessive water run-off resulting from extreme precipitation and poor drainage system quality in these regions. Studies reported that extreme rainfall might alter the water quality by flushing enteric pathogens into surface water, leading to higher diarrhea transmission [44,45]. Our study reports an association between extreme precipitation and diarrheal disease at a lag time of 0 months, which might be related to the period of pathogens’ incubation. A study reported the incubation period for bacillary dysentery and enteroviruses ranges from 2 to 10 days, which is support by our results [46].

Low and middle-income countries bear a disproportionate diarrhea burden. For instance, an Indian study suggested that lower humidity can elevate the risk of rotavirus diarrhea [47]. Another study in Vietnam reported that low relative humidity was associated with elevated risk of diarrhea [48]. Consistent with our results, low relative humidity increased the risk of diarrhea in the north and central regions of Surabaya. A study found that the effects of low relative humidity could persist for a lag time of up to 21 days [49]. However, we also found that extremely high relative humidity was significantly associated with the risk of diarrhea at a lag time of 0 and 1 months. In support of our findings, a study conducted in Vietnam showed that a high relative humidity level would elevate the risk of diarrhea after a lag time of 2 weeks [48]. A study conducted in a developed country in Asia, Singapore, reported that every 10% increase in relative humidity level is related to a 3% higher diarrhea risk (IRR: 1.030, 95% CI: 1.004–1.057) [50].The laboratory results from the studies identified that some causative agents of diarrhea, particularly rotavirus, are persistent for more than 10 days at room temperature with low and high relative humidity, but not at a moderate level [38,40,51]. Zhang et al. suggested that relative humidity is likely to promote synthesized effects rather than independent effects [41]. Therefore, further study is needed to investigate the nebulous relationship between relative humidity and diarrhea incidence in different areas and after different lag times.

This study observed a similar association between temperature-related or relative humidity-related diarrhea risk and lag time for the west, south, and east regions, and for the north and central regions in Surabaya, respectively (Figure 4 and Appendix A). There are numerous explanations for similarities in spatial risks. A prior study reported that the north and central regions are classified as inner city, while the west, south, and east regions are regarded as the city fringe [52]. The inner city is described as a region located within the central business area, mostly inhabited by individuals with a high level of education. On the other hand, the city fringe is characterized by proximity to industrial areas and surrounded by vast vacant lots and agricultural lands. A study conducted in central and north Surabaya reported that children infected with diarrhea are likely to have poor water, sanitation, and hygiene (WASH) practices. Meanwhile, people who live in the city fringe tend to use unimproved water sources for their domestic activities [53]. Unimproved water sources and sanitation are highly related to the transmission of diarrhea [54]. We believe that the incidence of diarrhea in Surabaya was related to WASH practices and exacerbated by food safety in the west, south, and east regions.

Our study provided an insight on risk disparity, which requires further investigation, particularly in different regions and after different lag times. Thus, it is necessary to understand the relationship between local climate and diarrhea incidence in different areas to signal early warnings. The early health warning system is a strong and reliable approach that has been proven to be an effective tool to provide the climate–health profile of a possible outbreak at least 2 weeks in advance [55]. Prior studies suggested that utilizing the available ambient and environment data that are closely related with the dynamics of disease (i.e., temperature, precipitation, relative humidity, etc.) to the fullest extent can significantly increase the chance to protect and enhance community health through simulation and prediction [55,56].

Even though this study is the first to identify the association between meteorological factors and diarrhea incidence in Surabaya, identifying potential meteorological risk factors responsible for diarrhea incidence, several limitations need to be addressed. First, this is an ecological study; thus, the risk was not approximated at the individual level. Second, meteorological factors are not the only risk factors for diarrhea. Our study did not consider demographic characteristics and living conditions that may modify the risk of diarrhea, such as the history of diseases, presence of personal dietary habits or refrigerators, knowledge of hygiene practices, and clean water coverage. In the present study, we were unable to incorporate the aforementioned information as variables due to the paucity of the data. We agree that adding those data is critical and omitting such conditions may affect the risk estimation. Moreover, a study in Indonesia reported that people will practice self-medication to treat diarrhea rather than seek for medical attention in the first place [57]. This behavior might cause underestimation of the quality of surveillance data.

This study recommends more scientific studies to understand the impact of extreme weather events and permit healthcare providers to cope with climate change. In addition, further study with different methods and more variables (environmental and socioeconomic factors) should be conducted to better understand the risks of diarrhea disease in local settings. This study provides scientific evidence for local authorities to improve diarrhea risk recognition. These methods can be applied to other diseases which are affected by the climate parameters. Conclusively, a weather-based early warning system and corresponding responses, such as prevention strategies and plans, are recommended to control endemic diarrhea.

## 5. Conclusions

Our study provided scientific evidence demonstrating that both extremely low and high temperatures were associated with elevated diarrhea risks in the west, south, and east regions of Surabaya. In addition, extremely high relative humidity and precipitation increased the risk of diarrhea in the north, central, and west regions of Surabaya after a lag time of 0 months. The findings of the present study could provide a reference for the development of a weather-based early warning notification; moreover, the temporal and spatial risk associations provide information for prevention strategies and planning to confront diarrhea in Surabaya. Therefore, we highly recommend that local health and meteorological bureaus collaborate to enhance community resilience against climate change. Integrating detailed health outcome and environmental data, regarding both temporal and spatial variables, can produce a robust early warning system that can lower the burden of diarrhea.

## Figures and Tables

**Figure 1 ijerph-20-02313-f001:**
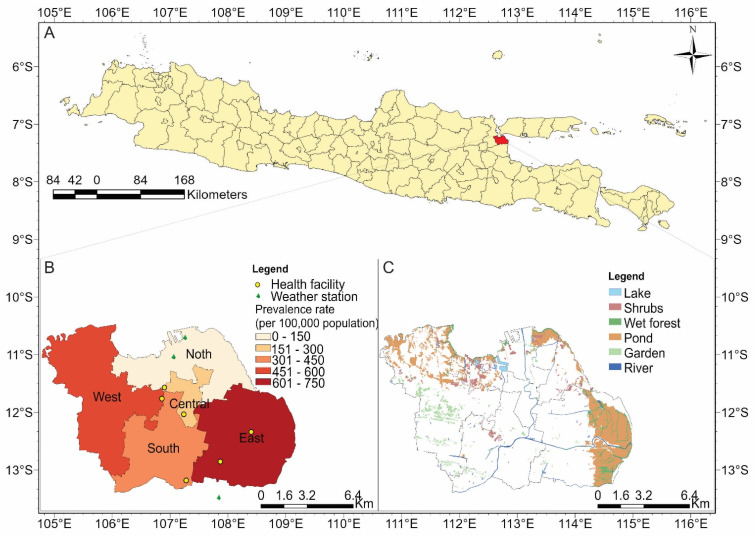
(**A**) Location of Surabaya on island of Java in Indonesia; (**B**) locations of weather stations and health facilities, and prevalence rate of diarrhea cases per 100,000 populations, from January 2018 to September 2020; (**C**) land use map for Surabaya.

**Figure 2 ijerph-20-02313-f002:**
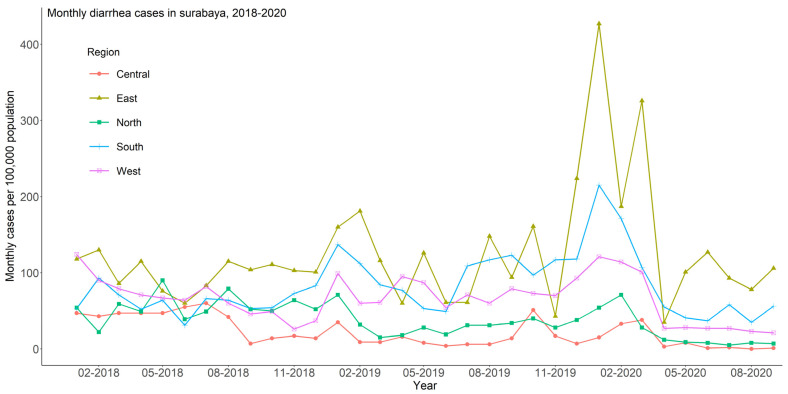
Monthly diarrhea cases in Surabaya from January 2018 to September 2020.

**Figure 3 ijerph-20-02313-f003:**
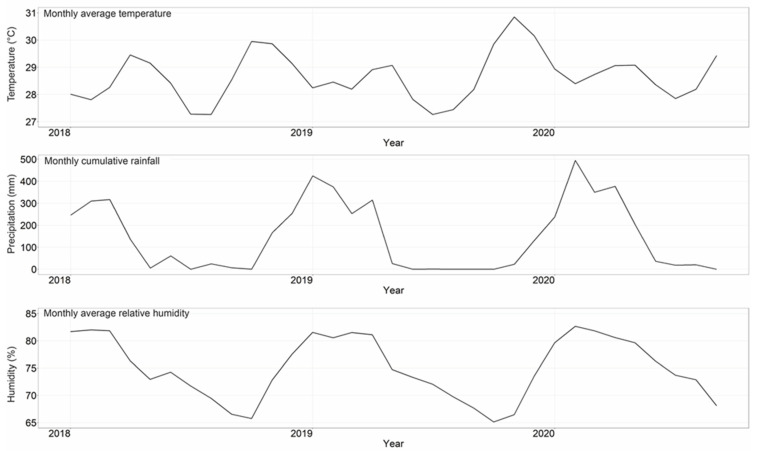
Monthly averages of temperature (°C), precipitation (mm), and relative humidity (%) in Surabaya from January 2018 to September 2020.

**Figure 4 ijerph-20-02313-f004:**
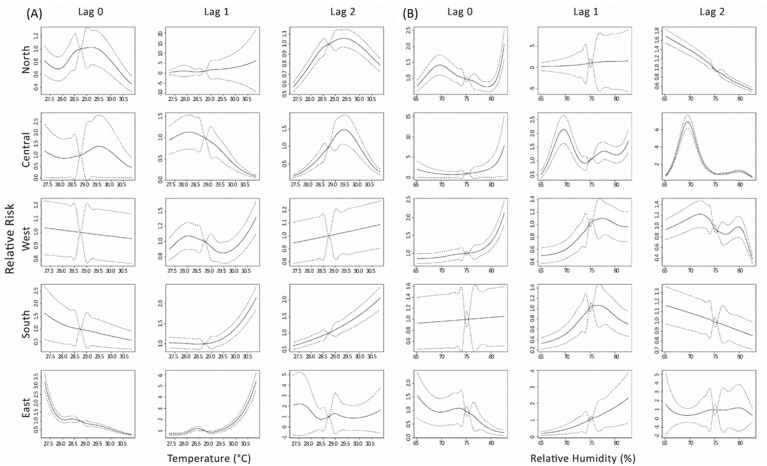
Lag associations between the relative risk of diarrhea and monthly (**A**) average temperature (°C), and (**B**) relative humidity (%) in Surabaya from January 2018 to September 2020.

**Table 1 ijerph-20-02313-t001:** Descriptive statistics of monthly averages of diarrhea cases and weather variables in Surabaya from January 2018 to September 2020.

	Sum of Cases	Mean (SD)	Min	5th	Q1	Q2	Q3	95th	Max
Diarrhea cases in Surabaya region									
North	1246	36.7 (22.6)	5	7	18	33	52	53.8	90
Central	723	21.9 (19.1)	0	1	7	15	42	52.6	60
West	2186	66.2 (29.3)	21	24.8	39	66	86	116	124
South	2725	82.6 (40.9)	31	36.2	53	71	109	151	215
East	4117	125 (78.2)	35	53.2	83	106	130	265	427
Weather measurements									
Temperature (°C)		28.7 (0.88)	27.3	27.3	28.1	28.5	29.2	30	30.9
Precipitation (mm)		155 (164)	0	0	6.2	102	268	442	443
Humidity (%)		75.2 (5.56)	65.1	66.3	71.6	74.5	80.7	82.2	82.8

**Table 2 ijerph-20-02313-t002:** Prevalence rate of diarrhea cases (per 100,000) by month in regions of Surabaya.

Region\Month	Jan	Feb	Mar	Apr	May	Jun	Jul	Aug	Sept	Oct	Nov	Dec	Average
North	6.61	4.62	3.77	2.92	4.69	2.44	3.14	4.36	3.43	4.98	5.10	4.98	4.25
Central	8.73	7.65	8.46	5.94	5.67	5.40	6.12	4.32	1.98	8.77	4.59	2.83	5.87
West	25.4	19.5	17.8	14.2	13.4	10.7	13.3	10.5	10.8	13.5	10.6	14.4	14.5
South	17.6	16.3	11.4	7.98	6.85	5.07	10.1	9.37	10.1	9.82	12.4	13.1	10.8
East	40.4	28.6	30.3	12.0	17.4	14.2	13.6	19.6	17.4	23.4	12.6	28.0	21.5

Rainy season: November to March; dry season: April to October.

## Data Availability

The datasets generated during and/or analyzed during the current study are not publicly available due to the original agreement.

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
