# Peer review of "Effects of Ambient Temperature, Relative Humidity, and Precipitation on Diarrhea Incidence in Surabaya"

_ijerph, 2023, doi:10.3390/ijerph20032313_

Round 1
Reviewer 1 Report
My compliments for this interesting article on this relevant public health topic. This study clearly adds to the current literature available on effects of meteorological conditions on diarrhea incidence. I see several major concerns mainly related to the methodology, which when addressed could further increase the added value of this work.
Major remarks
Lag periods. The lag time has been taken into account per month. I miss reasoning on why this time resolution and scale was chosen. It seems like both the surveillance data is available on a daily basis as well as the meteorological data. Considering pathophysiology of diarrhea, I can imagine a time resolution in weeks could potentially be more appropriate.
GAM. I miss argumentation how the number of df/knots was set for the spline, was penalization considered?
Multiple testing. Since for different lag times multiple correlated meteorological parameters were assessed, quite a high number of tests were performed. Why has multiple testing correction not been performed?
Surveillance data. It would be good if more information could be provided on the quality of the surveillance data. How many cases are missed? How many false positives?
Results. In the results and discussion section mainly the significant findings are highlighted. I miss reasoning on the connection overall between meteorological parameters and lag times. Eyeballing figure 4 seems like the shape of the associations is quite different between regions, and within regions between lag times. How can this be explained?
Other remarks
Line 251: ‘spatial risk similarity’ suggested to be described as ‘similarities in spatial risks’.
Supplemental information. I was not able to open this file so unfortunately I could not review it.
Author Response
Reviewer #1
My compliments for this interesting article on this relevant public health topic. This study clearly adds to the current literature available on effects of meteorological conditions on diarrhea incidence. I see several major concerns mainly related to the methodology, which when addressed could further increase the added value of this work.
Major remarks
Lag periods. The lag time has been taken into account per month. I miss reasoning on why this time resolution and scale was chosen. It seems like both the surveillance data is available on a daily basis as well as the meteorological data. Considering pathophysiology of diarrhea, I can imagine a time resolution in weeks could potentially be more appropriate.
Response: We thank the reviewer for pointing this out. We decided to use monthly resolution because the surveillance data we retrieved is only available in the monthly form. We have added the information regarding the outcome variables time resolution in Line 92 to 95, read:
“We retrieved monthly surveillance diarrhea data from January 2018 to September 2020 from six governmental public health centers admission in Surabaya, namely government public health centers of Puskesmas Dr. Soetomo, Asemrowo, Keputih, Siwalankerto, Rungkut, and Simomulyo (Figure. 1 B).”
To address and justify reviewer’s concern, we have added brief explanation in method section accordingly, which now read (Line 116 to 118):
“To match the temporal resolution between the exposure and outcome variables, we aggregated all region level daily meteorological data to monthly resolution to align it with health data.”
GAM. I miss argumentation how the number of df/knots was set for the spline, was penalization considered?
Response: Thank you for your comment which helps us to improve the readability of our manuscript. We have conducted sensitivity analyses to select the robust models according the lowest Akaike information criterion (AIC) by trying univariate and multivariate models and also changing the df value, 3-6 for meteorological variables and 3-7 for time. The final model of our study use 5df for meteorological variables and 7df for time. We have provided the details explanation in method section, please refer to Line 136 to 138:
“We tried df of 3 to 6 for meteorological variables and 3 to 7 for time, The model selection was based on the Akaike information criterion (AIC) value that a lower AIC value indicates a better model [1, 2].”
For your reference, we provide the table of RMSE and AIC which shows the same optimum combination for the best model. Please refer to Review Table 1.
Multiple testing. Since for different lag times multiple correlated meteorological parameters were assessed, quite a high number of tests were performed. Why has multiple testing correction not been performed?
Response: We thank the reviewer for pointing out this point. We followed the procedure from prior epidemiology studies using GAM [1, 3]. In GAM model, researchers performed correlation test (Spearman correlation and variance inflation factors) and sensitivity analysis (AIC and RMSE) to assess the model-fit to identify the optimized model setting.
Surveillance data. It would be good if more information could be provided on the quality of the surveillance data. How many cases are missed? How many false positives?
Response: We thank the reviewer for bringing this issues. We have added information regarding the patients’ privacy in data sources section. However, we cannot really say how many cases are missed or the false positives, because we collected the data in monthly count values.
Line 92 to 98: We retrieved monthly surveillance diarrhea data from January 2018 to September 2020 from six governmental public health centers admission in Surabaya, namely government public health centers of Puskesmas Dr. Soetomo, Asemrowo, Keputih, Siwalankerto, Rungkut, and Simomulyo (Figure. 1 B). Thus, we decided to define Surabaya City into those mentioned regions. Health outcome was collected as an aggregate count and all personal information of insured population had been striped, thus, the patient privacy is assured.”
Results. In the results and discussion section mainly the significant findings are highlighted. I miss reasoning on the connection overall between meteorological parameters and lag times. Eyeballing figure 4 seems like the shape of the associations is quite different between regions, and within regions between lag times. How can this be explained?
Response: We thank the reviewer for mentioning this issues which could increase the readability of our manuscript. We agree with reviewer that the previous version of this manuscript was lack of lag justification. However, we have worked really hard to revise and tailored our manuscript to accommodate the information related to lag effect. Please refer to Line 216 to 223 and 233 to 239 for the lag effects of temperature:
Line 216 to 223: “Our result showed that the risk of extreme low-related diarrheal disease was apparent at lag 0 months. Likewise, a study in Taiwan reported that low temperature associated with the risk of viral diarrhea up to lag of three weeks [4]. The extreme low temperature could enhance the replication and survival of viruses; hence, the incidence of viral diarrhea might be increased [5]. A study pointed out that rotavirus follows similar pattern as childhood viruses, measles, which are transmitted through the respiratory route in low temperature [6]. Another study found that rotavirus is more stable in low temperature compare to high temperature condition [7].”
Line 233 to 239: “Prior studies have reported the association between temperature and the risk of enteric disease at different lag ranging from 0 to 14 weeks [4, 8]. A study in Taiwan indicated that extreme hot temperature showed apparent effects on diarrhea disease at lag eight weeks on bacterial and all infectious diarrhea [4]. In line with our findings, we found the effects of high temperature on diarrhea incidence at lag one and two months and vary by regions. The prolonged effects of diarrheal could be caused by the inadequate aids to treat diarrhea and the re-infected cases [9].”
We also added the explanation regarding the lag effect of precipitation and relative humidity on diarrhea diseases. The revision can be seen on Line 246 to 249and Line 250 to 264:
Line 246 to 249: “Our study reports there was an association between extreme precipitation and diarrheal disease at lag 0 months, which might be related to the period of pathogens incubation. A study reported the incubation period for bacillary dysentery and enteroviruses is ranging from 2 to 10 days, which in support with our results [10].”
Line 250 to 264:” An Indian study suggested that lower humidity can elevate the risk of rotavirus diarrhea [11]. Consistent with our result, low relative humidity increased the risk of diarrhea in North and Central regions of Surabaya. A study found that the effects of low relative humidity could persist up to lag of 21 days [12]. However, we also found that extreme high relative humidity was significantly associated with the risk of diarrhea at lag of zero and one months. In support with our findings, a study conducted in Vietnam showed that high relative humidity level would elevate the risk of diarrhea at lag of four-weeks [13]. A study conducted in Singapore showed that every 10% increase in relative humidity level is related to a higher diarrhea risk [14].The laboratory results from the studies identified some causative agents of diarrhea, particularly rotavirus, are persistent for more than ten days at room temperature with low and high relative humidity, but not at moderate level [7, 15, 16]. Zhang et al. suggested that relative humidity is likely to promote synthesized effects rather than independent effects [8]. Therefore, further study is needed to investigate the nebulous relationship between relative humidity and diarrhea incidence in different areas and lag time.”
Other remarks
Line 251: ‘spatial risk similarity’ suggested to be described as ‘similarities in spatial risks’.
Response: Thank you for your comment. We have revised the sentences according to your suggestion, please refer to Line 268
Supplemental information. I was not able to open this file so unfortunately I could not review it.
Response: we are very sorry for the inconvenience, but we have uploaded all the files upon the submission.

Reviewer 2 Report
Dear Authors,
Thank you for submitting this paper that explores diarrhea incidence across a 3 year window in Surabaya. This is an interesting paper with some potential value to researchers and health services. There are some interesting points made in the review – though the real world use of the findings could be developed further.
There are some revisions required in order to consider this manuscript for publication. I have included specific feedback on the PDF document version of the manuscript, please find attached. Additionally, please address the following key areas when making revisions:
1. References and citations. Currently, no attempt has been made at formatting these to the stylistic requirements of the journal. Please revise all citations and references accordingly.
2. Methods. Be clear on how the records were taken and how representative of the region. Be clear on anonymity points.
3. Limitations. Consider whether the number of reported cases is a true reflection of the number of cases. Is there any evidence that some people with diarrhea will not seek medical help? If so, how might this affect your results? This needs to be considered in more detail.
4. P values. Make sure test outputs and p values are recorded more consistently in the work

Author Response
Reviewer #2
Dear Authors,
Thank you for submitting this paper that explores diarrhea incidence across a 3 year window in Surabaya. This is an interesting paper with some potential value to researchers and health services. There are some interesting points made in the review – though the real world use of the findings could be developed further.
There are some revisions required in order to consider this manuscript for publication. I have included specific feedback on the PDF document version of the manuscript, please find attached. Additionally, please address the following key areas when making revisions:
- References and citations. Currently, no attempt has been made at formatting these to the stylistic requirements of the journal. Please revise all citations and references accordingly.
Response: Thank you for your comment. We have downloaded and applied the IJERPH’s citation style on endnote, thus this is no longer to be an issue.
- Methods. Be clear on how the records were taken and how representative of the region. Be clear on anonymity points.
Response: We thank the reviewer for bringing this issues. We collected the data in person from major government public health centers, which cover and located in six regions of Surabaya. In addition, we have added information regarding the patients’ privacy in data sources section, please refer to Line 92 to 98:
“We retrieved monthly surveillance diarrhea data from January 2018 to September 2020 from six governmental public health centers admission in Surabaya, namely government public health centers of Puskesmas Dr. Soetomo, Asemrowo, Keputih, Siwalankerto, Rungkut, and Simomulyo (Figure. 1 B). Thus, we decided to define Surabaya City into those mentioned regions. Health outcome was collected as an aggregate count and all personal information of insured population had been striped, thus, the patient privacy is assured.”
- Limitations. Consider whether the number of reported cases is a true reflection of the number of cases. Is there any evidence that some people with diarrhea will not seek medical help? If so, how might this affect your results? This needs to be considered in more detail.
Response: Thank you for your valuable comment which helped us to improve the quality of our manuscript. A literature in reported that Indonesians tend to do self-medication to cure diarrhea rather than seek medical attention [18], which might lead to underestimate the quality of the surveillance system. However, surveillance data do not need to be perfect to be useful [17], thus, we believe that our data is representative to depict the actual condition and trend in study area, Surabaya, Indonesia. As per your suggestion, we have added this point to our limitation section, please refer to line 287 to 289, which now read:
“Moreover, a study in Indonesia reported that people will do self-medication to treat diarrhea rather than seek for medical attention in the first place [18]. This behavior might cause underestimate the quality of surveillance data.”
- P values. Make sure test outputs and p values are recorded more consistently in the work
Response: We thank the reviewer for pointing this out. We have tried to consistently display the p-value of less than 0.05. Please refer to Supplementary table S1 and table S2
Comments from PDF file:
- Line 30: Why the change in font here?
Response: Thank you for your comment. We have revised it accordingly. Please refer to Line 30.
- Line 41: This study suggests the environmental and health sectors co-develop a weather-based early warning system for implementing local sanitation practices to respond to increasing risks of infectious diseases. Explain this last point further and check spacing here.
Response: Thank you for your comment. We have revised the sentence to increase the clarity of the manuscript, which now read (Line 38 to 41):
“This study suggests the local environmental and health sectors to co-develop a weather-based early warning system and improve local sanitation practices as prevention measures in response to increasing risks of infectious diseases.”
- Provide further keywords
Response: Thank you for your comment. We have updated the keywords to Diarrhea; Generalized additive model; Meteorological variables; Surabaya.
- Please make sure citations are formatted as per the MDPI guidelines. They should be reformatted as numbers.
Response: Thank you for your comment. We have downloaded and applied the IJERPH’s citation style on endnote, thus this is no longer to be an issue.
- Line 47: deaths
Response: Thank you for your comment. We have revised it accordingly. Please refer to Line 46.
- Line 58: the full scientific name needs to be provided on the first mention.
Response: Thank you for your comment. We have revised it accordingly. Please refer to Line 56.
- Line 61: good but explain how here
Response: Thank you for your suggestion. We have tried to reason how higher humidity affects the replication of bacteria and protozoa. Please refer to Line 58 and 61:
“Higher humidity also affects the replication of bacteria and protozoa since higher humidity will promotes the formation of biofilm on the surface; which will provide protection and favorable environment for microorganisms [10-12].”
- Line 83: inhabitants
Response: Thank you for your comment. We have revised it accordingly. Please refer to Line 81.
- Line 98: What proportion of inhabitants are covered by these areas? Is it the 96% below? Be clear in wording
Response: Thank you for your comment. Those health care facilities are supposed to cover the whole inhabitants (100%) in this study areas. We have deleted the part that explained about Indonesia Health Social Security Agency (Badan Penyelenggara Jaminan Sosial Kesehatan, BPJS) to clear the confusion.
- Line 103: Be a bit clearer on how the data were extracted. Was it all the cases that involved diarrhea? What if there were multiple symptoms and diarrhea was just one of them? How were data kept anonymous?
Response: We thank the reviewer for bringing this issues. We collected the data in person from major government public health centers, which cover and located in six regions of Surabaya. This surveillance data include all kind of diarrhea (infectious and non-infectious diarrhea). Clinician or health expert will record the symptoms with diarrhea simultaneously with other visible symptoms. In addition, we have added information regarding the patients’ privacy in data sources section, please refer to Line 92 to 98:
“We retrieved monthly surveillance diarrhea data from January 2018 to September 2020 from six governmental public health centers admission in Surabaya, namely government public health centers of Puskesmas Dr. Soetomo, Asemrowo, Keputih, Siwalankerto, Rungkut, and Simomulyo (Figure. 1 B). Thus, we decided to define Surabaya City into those mentioned regions. Health outcome was collected as an aggregate count and all personal information of insured population had been striped, thus, the patient privacy is assured.”
- check subsubheading formatting
Response: Thank you for your comment. We have revised it accordingly.
- Line 113: no a percentage. A percentage is out of 100
Response: Thank you for your comment. We have revised it accordingly. Please refer to Line 110.
- Line 125: follows
Response: Thank you for your comment. We have revised it accordingly. Please refer to Line 123.
- Line 132: control for seasonal
Response: Thank you for your comment. We have revised it accordingly. Please refer to Line 130.
- Figure 2: these are difficult to differentiate. use colour to show the different regions; y-axis information; make the exact month clearer
Response: Thank you for your comment. We have replotted the figure using color to differentiate the region, revised the y-axis title, and made the x-axis clearer compare to previous version. Please refer to Figure 2.
- Line 175: provide the r and p value here
Response: Thank you for your comment. We have revised it accordingly.
- Line 220: explain this further
Response: Thank you for your comment. We have added further explanation regarding how rotavirus works better in low temperature. Favorable environment is a survivability key driver for viruses, such as rotavirus and measles. A study found that rotavirus is more stable in low temperature and it will lose the effectivity in higher temperature (37°C)[7]. A study in support with that statement reported that rotavirus has similar transmission patterns as measles virus, which is which are transmitted through the respiratory route in low temperature [6]. Please refer to line 218 and 223:
“The extreme low temperature could enhance the replication and survival of viruses; hence, the incidence of viral diarrhea might be increased [5]. A study pointed out that rotavirus follows similar pattern as childhood viruses, measles, which are transmitted through the respiratory route in low temperature [6]. Another study found that rotavirus is more stable in low temperature compare to high temperature condition [7]..”
- Line 227: spoil
Response: Thank you for your comment. We have revised it accordingly. Please refer to Line 230.
- References: No attempt has been made at preparing the references to MDPI style. Please revise the references carefully as per the author guidelines.
Response: Thank you for your comment. We have downloaded and applied the IJERPH’s citation style on endnote, thus this is no longer to be an issue.
- 1: Check the punctuation carefully.
Response: Thank you for your comment. We have removed the mentioned citation, thus this is no longer to be an issue.
- Ref 4: please check position of the year
Response: Thank you for your comment. We have followed the IJERPH’s citation style, thus this is no longer to be an issue.
- Ref 7: Boithias L, Choisy M, Souliyaseng N, Jourdren M, Quet F, Buisson Y, Thammahacksa C, Silvera N, Latsachack K, Sengtaheuanghoung O, Pierret A, Rochelle-Newall E, Becerra S, Ribolzi O (2016) Hydrological Regime and Water Shortage as Drivers of the Seasonal Incidence of Diarrheal Diseases in a Tropical Montane Environment. PLOS Neglected Tropical Diseases 10 (12):e0005195. doi:10.1371/journal.pntd.0005195. Pages are missing here
Response: Thank you for your comment. We have checked the page number and followed the journal citation style. e0005195 is the journal’s page number.
- Ref 33: Rosenberg A, Weinberger M, Paz S, Valinsky L, Agmon V, Peretz C (2018) Ambient temperature and age-related notified Campylobacter infection in Israel: A 12-year time series study. Environmental Research 164:539-545. 382 doi:https://doi.org/10.1016/j.envres.2018.03.017. Scientific names should be in italics.
Response: Thank you for your comment. We have made the scientific name italic.
- Ref 37: U. S. Department of Health Human Services aCfD, Control Prevention, (2011) Principles of epidemiology in public health practice: an introduction to applied epidemiology and biostatistics. Reference seems incomplete
Response: Thank you for your comment. We made a mistake by input this citation as journal article when it is should be a book. We have updated the citation accordingly, which now: U. S. Department of Health Human Services, a. C. f. D., Control Prevention,, Principles of epidemiology in public health practice : an introduction to applied epidemiology and biostatistics. Third edition ed.; U.S. Department of Health and Human Services, Centers for Disease Control and Prevention: Atlanta, GA, 2011.
- Ref 40: why the year twice
Response: Thank you for your comment. We have followed the IJERPH’s citation style, thus this is no longer to be an issue.

Round 2
Reviewer 2 Report
Dear Authors,
Many thanks for submitting this revised version of the manuscript for review. You have taken into account the feedback provided on the initial review of the paper. You have also shown clearly where changes have been made to the work. The developments to the manuscript have resulted in a more robust paper overall.